# Vision-Related Parameters Affecting Stereopsis after Retinal Detachment Surgery

**DOI:** 10.3390/jcm12041527

**Published:** 2023-02-15

**Authors:** Fumiki Okamoto, Tomoya Murakami, Shohei Morikawa, Yoshimi Sugiura, Takahiro Hiraoka, Tetsuro Oshika

**Affiliations:** Department of Ophthalmology, Faculty of Medicine, University of Tsukuba, 1-1-1 Tennoudai, Tsukuba 305-8575, Japan

**Keywords:** aniseikonia, contrast sensitivity, metamorphopsia, retinal detachment, stereopsis

## Abstract

Even after successful surgery, the stereopsis of retinal detachment (RD) patients is inferior to that of normal subjects. However, it is unclear which visual dysfunction in the affected eye is responsible for the postoperative stereopsis impairment. This study included 127 patients after successful surgery for unilateral RD. Stereopsis, best-corrected visual acuity (BCVA), severity of metamorphopsia, letter contrast sensitivity and amount of aniseikonia were examined at 6-month postoperatively. Stereopsis was assessed using the Titmus Stereo Test (TST) and TNO stereotest (TNO). Postoperative stereopsis (log) in patients with RD were 2.09 ± 0.46 in the TST and 2.56 ± 0.62 in the TNO. Multivariate analysis with stepwise regression revealed postoperative TST was associated with BCVA, and TNO was associated with BCVA, letter contrast sensitivity, metamorphopsia and absolute values of aniseikonia. In a subgroup analysis that selected those with more impaired stereopsis, postoperative TST was associated with BCVA (*p* < 0.001), and TNO was associated with letter contrast sensitivity (*p* < 0.005) and absolute values of aniseikonia (*p* < 0.05) by multivariate analysis. Deterioration of stereopsis after RD surgery was affected by a variety of visual dysfunctions. The TST was affected by visual acuity, while the TNO was affected by contrast sensitivity and aniseikonia.

## 1. Introduction

Stereopsis is the most sophisticated visual functions, in which depth of field is recognized by the disparity between the images formed by the two eyes. Strabismus is a typical disorder in which stereopsis is impaired due to disruption of binocular coordination. However, stereopsis may also be impaired by visual disturbance in one eye. The most common visual function factor affecting stereopsis is visual acuity. Previous reports have shown that stereopsis is impaired when visual acuity in one eye is reduced [1,2,3,4,5]. In addition, various factors including aniseikonia [6,7,8], pupil size [8,9], eye dominance [6,10] and accommodation [5,11] have been reported to affect stereopsis.

Several studies have investigated the disturbance of stereopsis in patients after retinal detachment (RD) [12,13] and have found that even if the surgery is successful, stereopsis of RD patients is inferior to that of normal subjects [12]. It has also been reported that stereopsis after RD patients affects their vision-related quality of life (VR-QOL), especially driving [14]. On the other hand, patients after RD have not only blurred vision but also various visual disturbance, such as severity of metamorphopsia [15,16,17,18,19,20,21], amount of aniseikonia [22,23,24,25] and decreased contrast sensitivity [26,27]. According to previous reports, postoperative metamorphopsia is caused by retinal shift [15,17,18,19], development of epiretinal membrane (ERM) [15,21] and outer retinal folds [21]. Appearance of aniseikonia is caused by preoperative macula-off detachment [22,25], cystoid macular edema [22,23], development of ERM [23]. and extent of preoperative RD [23]. Decrease in contrast sensitivity is caused by macula-off RD [27], extent of retinal tears [26]. It has been reported that the cause of impaired stereopsis in ERM patients is aniseikonia [28], and the cause of impaired stereopsis in macular hole (MH) patients is decreased presence of aniseikonia and contrast sensitivity [29]. However, the effects of visual dysfunction including visual acuity, contrast sensitivity, severity of metamorphopsia and amount of aniseikonia on stereopsis in patients after RD surgery have not been clarified (Figure 1). The purpose of the study was to investigate stereopsis and other visual function factors in patients after unilateral RD, and to identify visual function parameters that affect stereopsis.

## 2. Materials and Methods

Patients who underwent unilateral retinal detachment surgery at the University of Tsukuba Hospital and were followed up for at least 6 months after surgery were included. Exclusion criteria included eyes with (1) a history of vitrectomy, (2) myopia more than −10.0 diopters, (3) moderate and severe cataract (more than grade 3 cortical opacity or nuclear sclerosis and (4) unconventional RD due to proliferative vitreoretinopathy, giant retinal tears, ocular trauma and macular holes. This study was conducted in accordance with the Declaration of Helsinki and was approved by the Institutional Review Board of the University of Tsukuba Hospital. Before participating in this study, all patients gave informed consent after being informed of the nature of the study.

### 2.1. Assessments

Stereopsis, best-corrected visual acuity (BCVA), letter contrast sensitivity, metamorphopsia and aniseikonia were examined at 6-month postoperatively. We converted the BCVA measured on the Landolt chart to the logarithm of the minimum angle of resolution (logMAR) and used it for subsequent analysis.

### 2.2. Stereopsis

Stereopsis was evaluated using the Titmus stereo test (TST) and the TNO stereotest (TNO). TST uses polarization filters, and TNO stereotest uses red–green filters to separate the two eyes and measure stereopsis. These tests were performed at a standard viewing distance of 40 cm under correction. To ensure that the TSTs were not using monocular cues, we flipped the stereo target and asked if the target was forward or backward on the page to see the response. The results for TST and TNO are expressed in “seconds of arc”, but for statistical evaluation these values were converted to logarithms [30,31].

### 2.3. Contrast Sensitivity

Letter contrast sensitivity was assessed using the CSV-1000LV chart (Vector Vision, Greenville, OH, USA). In this test, the contrast was variable, and the spatial frequency was constant. The chart has 8 pairs of 3 letters with the same contrast, for a total of 24 letters. Each triplet decreased in contrast in turn. If all the letters were unreadable, the score was 0, and the maximum score was 24 [32].

### 2.4. Metamorphopsia

Metamorphopsia was evaluated with M-CHARTS (Inami Co., Tokyo, Japan). M-CHARTS consists of dotted lines from 0.2 to 2.0 degrees of visual angle. If the patient initially perceives a straight line at 0 degrees as a curve, he or she is considered to have metamorphopsia. Then, the spacing of the dotted lines is varied from fine to coarse. When the patient perceives the dotted line as a straight line, the visual angle of the dotted line is the patient’s metamorphopsia score [33,34]. Both vertical and horizontal meridians were evaluated, and the mean values were used for data analysis.

### 2.5. Aniseikonia

Aniseikonia was measured by the new aniseikonia test (NAT) (Handaya, Tokyo, Japan). The NAT is capable of measuring the aniseikonia in the range of ±24%. [35]. The NAT test consists of a pair of red and green semicircles of target size 4 cm, with two semicircles of different sizes indexed in 1% increments. When subjects wore the red/green glasses, the two semicircles appeared to be separated into one each. The difference in the actual size of the two semicircles when the subject selected a pair in which they appear to be the same size represents the amount of aniseikonia of the subject. We measured the vertical and horizontal meridians and used their average values for data analysis. Patients with logMAR BCVA of >1.0 were excluded since it was difficult for them to perceive the half-moon due to suppression [35].

### 2.6. Retinal Detachment Surgery

Three experienced vitreoretinal surgeries (F.O., Y.S., T.H.) were performed under sub-Tenon local or general anesthesia. We performed a 25-gauge vitrectomy with dissection the vitreous around the breaks, drainage of subretinal fluid, fluid-air or fluid-20% sulfur hexafluoride (SF_6_) gas exchange, and endlaser photocoagulation for retinal breaks. There was no additional scleral buckling in any of the vitrectomy groups. The buckling surgery consisted of circumferential silicone sponge buckling (no. 506; MIRA, Walthom, MA, USA), external drainage and cryopexy. The encircling was performed with a silicone band (no. 240; MIRA) or a silicone sponge (no. 506G; MIRA). Patients treated with air/SF_6_ injection were instructed to maintain the facedown position for 1–3 days.

### 2.7. Statistical Analysis

The Wilcoxon signed-ranks test was performed to compare preoperative and postoperative results. The Mann–Whitney *U* test was used to compare each visual function between macula-on RD and macula-off RD patients. Associations between TST and TNO values and the other parameters of visual function, including BCVA, letter contrast sensitivity, severity of metamorphopsia, and amount of aniseikonia were examined by the Spearman rank correlation test. Multivariate analysis with stepwise regression was performed to evaluate the relationship between stereopsis and other vision-related parameters. All tests were considered statistically significant if *p* was < 0.05. Analyses were performed using StatView (version 5.0, SAS Inc., Cary, NC, USA).

## 3. Results

### 3.1. Clinical Features and Visual Function in Patients with Retinal Detachment

A total of 127 patients (82 males and 45 females; mean age [±SD], 55.3 ± 11.1 years) were included in the study. Table 1 shows the clinical features and visual functions in patients with RD. Preoperative BCVA was 0.54 ± 0.78 and postoperative BCVA was 0.04 ± 0.18, with significant improvement after surgery (*p* < 0.0001). All patients had unilateral RD and surgery only in the affected eye during the follow-up period and retinal reattachment was attained at primary operation. Sixty-eight eyes were macula-on RD, and 59 eyes were macula-off RD. Among the 127 RD patients, 23 were performed scleral buckling surgery and 104 were performed vitrectomy. Among the 104 eyes underwent vitrectomy, 9 eyes were pseudophakic and 95 eyes were phakic. Of the 95 eyes, 81 eyes received combined cataract surgery and vitrectomy and 14 eyes underwent lens sparing vitrectomy. All 14 eyes with lens sparing vitrectomy were under 50 years of age (43.7 ± 3.5 years), and cataract did not progress during the 6-month observation period. No significant intra- and postoperative complications including endophthalmitis, subretinal hemorrhage, and choroidal detachment were observed.

### 3.2. Relationship between Titmus Stereo Test and the Other Visual Functions after Surgery

Postoperative stereopsis in the TST showed significant correlation with preoperative and postoperative BCVA, letter contrast sensitivity, metamorphopsia score and absolute values of aniseikonia (Table 2). Multivariate analysis revealed TST was associated with postoperative BCVA (*p* < 0.0001), but not with the other vision-related parameters.

### 3.3. Relationship between TNO Stereotest and Other Visual Functions after Surgery

Postoperative stereopsis in the TNO showed significant correlation with preoperative and postoperative BCVA, letter contrast sensitivity, metamorphopsia score and absolute values of aniseikonia (Table 3). Multivariate analysis revealed TNO was associated with postoperative BCVA (*p* < 0.01), letter contrast sensitivity (*p* < 0.0001), metamorphopsia score (*p* < 0.05) and absolute values of aniseikonia (*p* < 0.005).

### 3.4. Subgroup Analysis of Patients after Retinal Detachment Surgery with Impaired Stereopsis

We conducted a subgroup analysis by selecting only patients with impaired stereopsis, namely, (1) 55 patients with TST value >2.0 who could not answer circle 5 or more correctly, (2) 77 patients with TNO value > 2.3, and 79 patients who could only read up to plate V were included. As a result, postoperative stereopsis in the TST was associated with only postoperative BCVA (*p* < 0.001, Figure 2a), and TNO was associated with letter contrast sensitivity (*p* < 0.005) and absolute values of aniseikonia (*p* < 0.05, Figure 2b,c) by multivariate analysis.

### 3.5. Subgroup Analysis with Preoperative Macular Status

We divided the patients into preoperative macula-on and macula-off RD and analyzed the visual function factors affecting stereopsis. Table 4 shows the clinical features and visual functions in macula-on and macula-off RD. The macula-off RD group was significantly worse than the macula-on RD group in all visual functions including stereopsis. In the macula-off group, postoperative TST and TNO were associated with postoperative BCVA (*p* < 0.001) by multivariate analysis. In the macula-on group, postoperative TST was associated with postoperative BCVA (*p* < 0.05) and absolute values of aniseikonia (*p* < 0.05), and postoperative TNO was associated with letter contrast sensitivity (*p* < 0.001) and absolute values of aniseikonia (*p* < 0.001) by multivariate analysis.

## 4. Discussion

Postoperative stereopsis after RD surgery were 2.09 in the TST and 2.56 in the TNO. It has been reported that stereopsis is more impaired after RD surgery than in normal subjects [12,36]. Two previous studies quantified stereopsis after RD surgery, and the results were 200 s of arc (log 2.30) [14] and 269 s of arc (log 2.43) [12] in TST, respectively. Our TST result (2.09) was better than previous reports, possibly because the percentage of patients with macula-off RD was lower in our report (46.5%) than in previous report (62.5%) [14]. Several studies have investigated stereopsis even after treatment for retinal diseases such as unilateral ERM, MH and branch retinal vein occlusion (BRVO). Stereopsis after treatment in ERM patients were 2.19 in the TST and 2.63 in the TNO [30], MH were 2.2 in the TST and 2.4 in the TNO [31], and BRVO were 2.06 in the TST and 2.32 in the TNO [37]. These findings showed that the degree of impaired stereopsis after RD surgery was similar to that of other unilateral retinal diseases. However, in this study, postoperative TST and TNO of macula-off RD were 2.31 and 2.85, respectively, worse than those of other retinal diseases. Postoperative stereopsis is associated with quality of life, especially driving, and patients after macula-off RD need special attention [14].

Postoperative TST was significantly associated with all visual function parameters including BCVA, contrast sensitivity, severity of metamorphopsia and amount of aniseikonia. By multivariate analysis, TST was associated with only postoperative BCVA. TST and TNO have also been reported to be associated with visual acuity in other retinal diseases such as ERM [28], MH [29], and BRVO [37]. Stereopsis is influenced by visual acuity [1,2,3,4,5]. Healthy subjects who can identify TST circle 9 can only identify TST circle 3 when the visual acuity of one eye is reduced to 20/200 by an experimental convex lens. In addition, the degree of visual impairment and stereopsis have been shown to be significantly correlated [4]. Postoperative mean BCVA was 0.04 (logMAR) in this study, which is relatively good, but the TST was 2.09 (log), which is only enough to recognize circle 5. In Macular-off RD patients alone, BCVA was 0.187, and TST was even worse, around circle 3. Postoperative TNO was also associated with all visual function parameters by both single correlation and multiple regression. Impairment of stereopsis may be due to a combination of factors of visual function.

We performed a subgroup analysis of 77 patients with significantly impaired stereopsis with TST values > 2.0 or TNO values > 2.3. Results showed that TST was significantly associated with postoperative BCVA and TNO with letter contrast sensitivity and aniseikonia in multivariate analysis. Both visual acuity and contrast sensitivity are visual function factors that represent morphosensory perception, but contrast sensitivity is considered to be a more sensitive indicator of morphosensory perception than visual acuity. In recent years, the importance of contrast sensitivity in retinal diseases has been widely reported. Contrast sensitivity in patients with mild ERM [32], macula-on RD [26], and vitreous floaters [38] is reduced even when the visual acuity is good. Not visual acuity but contrast sensitivity showed correlation with the VR-QOL in patients with RD [26], proliferative diabetic retinopathy and diabetic macular edema [39]. In addition, previous study reported that contrast sensitivity is one of the causes of impaired stereopsis in MH patients [29]. Thus, stereopsis is probably impaired even if contrast sensitivity is disturbed in only one eye.

Several reports reported that the amount of aniseikonia affected stereopsis [6,7,8]. TNO in RD patients with impaired stereopsis was associated with aniseikonia in this study. Brooks et al. reported that experimentally produced 1 diopter of anisometropia (about 1% aniseikonia) in healthy subjects resulted disruption of stereopsis [40]. Lovasik et al. reported that TST decreased from 40 to 200 s of arc when 3% anisometropia was applied to one eye [8]. The mean absolute value of aniseikonia after RD surgery in this study was 2.4%. Therefore, this level of aniseikonia suggests that patients after RD sugrery may be sufficiently impaired in stereopsis.

Our study had several limitations. We evaluated the patients for 6 months postoperatively. RD patients have been shown to have better visual acuity 1–5 years postoperatively [41,42], and aniseikonia continues to improve 7–45 months postoperatively [43]. Other factors known to affect stereopsis include pupil size [8,9], eye dominance [6,10] and accommodation [5,11], but these factors were not assessed in this study. However, the effects of pupil size and eye dominance on stereopsis seem to be slight. It has been reported that when pupil size changed from 1 mm to 6 mm, the decrease in TST value was 0.18 (log) [8], and even if the dominant eye changed, the decrease in TST value was 0.2 (log) [6]. Future studies with longer follow-up duration and incorporation of other factors will further improve our understanding of stereopsis inpatients with RD. Since the results of our study are retrospective, these results should be considered with caution. Future prospective cohort studies are needed to confirm these results.

In conclusion, our study demonstrated that stereopsis impairment in patients after retinal detachment surgery was associated with various visual dysfunctions. Titmus Stereo Test results were influenced by visual acuity, and the TNO stereotest results by contrast sensitivity and aniseikonia.

## Figures and Tables

**Figure 1 jcm-12-01527-f001:**
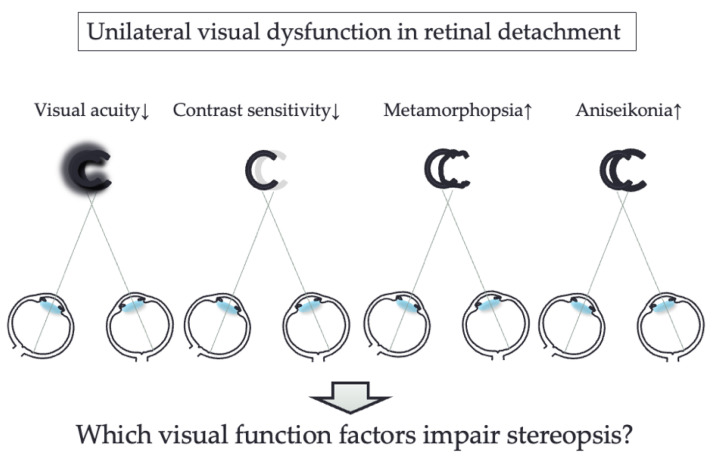
Conceptual illustration of the effect of impairment of visual function factors in one eye on stereopsis.

**Figure 2 jcm-12-01527-f002:**
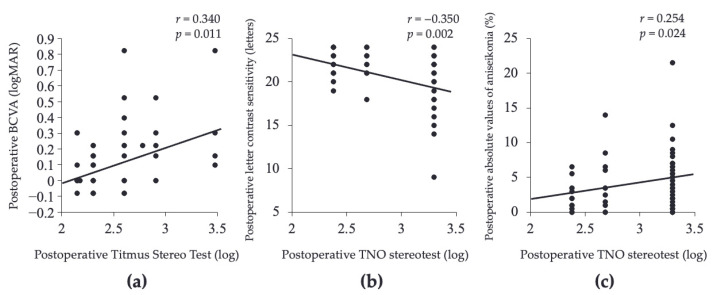
Association between stereopsis and visual function factors in patients with impaired stereopsis after retinal detachment surgery. The TST was associated with postoperative BCVA (**a**), and the TNO was associated with letter contrast sensitivity (**b**) and absolute values of aniseikonia (**c**).

**Table 1 jcm-12-01527-t001:** Clinical features and visual functions in patients with retinal detachment (RD).

Number of Eyes	127
Age (years)	55.3 ± 11.1
Sex (men/women)	82/45
Preoperative macular status (macula-on/macula-off)	68/59
Preoperative BCVA (logMAR)	0.54 ± 0.78
Postoperative BCVA (logMAR)	0.04 ± 0.18
Postoperative TST (log)	2.09 ± 0.46
Postoperative TNO stereotest (log)	2.56 ± 0.62
Postoperative letter contrast sensitivity (letters)	21.9 ± 2.3
Postoperative metamorphopsia (°)	0.28 ± 0.43
Postoperative absolute values of aniseikonia (%)	2.4 ± 3.5

Values are presented as the mean ± standard deviation. BCVA = best-corrected visual acuity, LogMAR = logarithm of the minimum angle of resolution, TST = Titmus Stereo Test.

**Table 2 jcm-12-01527-t002:** Correlation between Titmus Stereo Test (TST) results and other visual functions after retinal detachment surgery.

Parameters of Visual Function	Postoperative TST	
	*r*	*p*
Preoperative best-corrected visual acuity	0.417	<0.0001 *
Postoperative best-corrected visual acuity	0.565	<0.0001 *
Postoperative letter contrast sensitivity	−0.252	<0.005 *
Postoperative metamorphopsia	0.373	<0.0001 *
Postoperative absolute values of aniseikonia	0.316	<0.005 *

* Significant correlation between parameters (Spearman’s rank correlation test).

**Table 3 jcm-12-01527-t003:** Correlation between TNO stereotest results and other visual functions after retinal detachment surgery.

Parameters of Visual Function	Postoperative TNO Stereotest	
	*r*	*p*
Preoperative best-corrected visual acuity	0.371	<0.0001 *
Postoperative best-corrected visual acuity	0.575	<0.0001 *
Postoperative letter contrast sensitivity	−0.382	<0.0001 *
Postoperative metamorphopsia	0.483	<0.0001 *
Postoperative absolute values of aniseikonia	0.383	<0.0001 *

* Significant correlation between parameters (Spearman’s rank correlation test).

**Table 4 jcm-12-01527-t004:** Clinical features and visual functions in patients with macula-on and macula-off retinal detachment (RD).

	Macula-On RD	Macula-Off RD	*p* Values
Number of eyes	68	59	
Age (years)	52.4 ± 10.7	58.7 ± 10.7	0.001
Sex (men/women)	43/25	39/20	0.88
Preoperative BCVA (logMAR)	0.10 ± 0.49	1.05 ± 0.75	<0.001
Postoperative BCVA (logMAR)	−0.04 ± 0.08	0.13 ± 0.21	<0.001
Postoperative TST (log)	1.89 ± 0.33	2.31 ± 0.48	<0.001
Postoperative TNO stereotest (log)	2.30 ± 0.58	2.85 ± 0.54	<0.001
Postoperative letter contrast sensitivity (letters)	22.2 ± 2.5	21.6 ± 2.2	0.016
Postoperative metamorphopsia (º)	0.11 ± 0.31	0.48 ± 0.46	<0.001
Postoperative absolute values of aniseikonia (%)	1.2 ± 1.9	3.8 ± 4.3	<0.001

Values are presented as the mean ± standard deviation. BCVA = best-corrected visual acuity, LogMAR = logarithm of the minimum angle of resolution, TST = Titmus Stereo Test.

## Data Availability

Study data are available from the corresponding author on reasonable request.

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
