# Peer review of "Vision-Related Parameters Affecting Stereopsis after Retinal Detachment Surgery"

_jcm, 2023, doi:10.3390/jcm12041527_

Round 1
Reviewer 1 Report (Previous Reviewer 1)
The authors solved the proposed comment and incorporated all the changes in the manuscript.
the manuscript was improved since the previous version
Thanks to the authors
Reviewer 2 Report (Previous Reviewer 2)
Good revision of the manuscript.
This manuscript is a resubmission of an earlier submission. The following is a list of the peer review reports and author responses from that submission.
Round 1
Reviewer 1 Report
The authors study the event that stereopsis after retinal detachment surgery is inferior to normal subjects and the found that deterioration of stereopsis after RD surgery was affected by a variety of visual dysfunction and Titmus was affected by visual acuity, while TNO was affected by contrast sensitivity and aniseikonia.
The topic os the manuscript is of the interest of the retina surgeon and optometrist scientific community. The research is well conducted but some issues must be solved prior to continue with the publication process.
In the introduction must be clear state the difference between the concept presented in the Figure 1 along the text in the manuscript.
The figure legend of the figure 1 should be resume and include only the relevant information
Include at the beginning of the material and method the design of the research
Patients and age at the beginning of the results
Differentiate the inclusion and exclusion criteria in two different sentences with numbers of order
Did you think 6 months is correct date to revise the variables?
Explain the difference between both stereopsis methods
Include validation references in the variables measure
In the results present a table or graph or written results in where the results were compared before and after the surgery with the p value, not only the relationship with a correlation value
The unit of the stereopsis are commonly degrees, why are you putting in other units?
If some previous studies report de previous and posterior stereopsis to a RD surgery it is important to introduce a simple table with these results at the discussion
In the subgroups and visual acuity and contrast sensitivity paragraph some part were difficult to read, please revise this section and rewrite to improve the comprehension.
The conclusion of the study is missing at the end of the manuscript, please include two three sentences that resume the relevant findings of the manuscript.
In the references try, when possible, to include only recent and updated references after 2010 and also include only indexed journal.
Reviewer 2 Report
Thank you for the opportunity to review this article assessing visual stereopsis after retinal detachment.
Although addressing several important aspects of visual recovery after surgical repair for retinal detachment, this article presents several issues that need to be addressed before considering the submission.
- Please review the article for english and spelling mistakes
- Please report on the phakic status of the patients and analyze the functional parameters also in subgroups. It would be important to identify those 23 eyes operated by vitrectomy alone and mention if and when they developed a cataract after the vitrectomy. This can be a confounding bias.
- Another aspect to consider is the type of retinal detachment, and the status of the fovea. Event though the absolute numbers of macula on/off eyes are reported in the results, anaylsis of the functional parameters might be more meaningful if carried out separating the results.
- Please clearly mention the retrospective nature of your study as one of the most important limitations in the discussion section.